# Illumination-Based Color Reconstruction for the Dynamic Vision Sensor

**DOI:** 10.3390/s23198327

**Published:** 2023-10-09

**Authors:** Khen Cohen, Omer Hershko, Homer Levy, David Mendlovic, Dan Raviv

**Affiliations:** The Faculty of Engineering, Department of Physical Electronics, Tel Aviv University, Tel Aviv 69978, Israel; khencohen@mail.tau.ac.il (K.C.);

**Keywords:** computational photography, dynamic vision sensor, color reconstruction, active illumination

## Abstract

This work demonstrates a novel, state-of-the-art method to reconstruct colored images via the dynamic vision sensor (DVS). The DVS is an image sensor that indicates only a binary change in brightness, with no information about the captured wavelength (color) or intensity level. However, the reconstruction of the scene’s color could be essential for many tasks in computer vision and DVS. We present a novel method for reconstructing a full spatial resolution, colored image utilizing the DVS and an active colored light source. We analyze the DVS response and present two reconstruction algorithms: linear-based and convolutional-neural-network-based. Our two presented methods reconstruct the colored image with high quality, and they do not suffer from any spatial resolution degradation as other methods. In addition, we demonstrate the robustness of our algorithm to changes in environmental conditions, such as illumination and distance. Finally, compared with previous works, we show how we reach the state-of-the-art results. We share our code on GitHub.

## 1. Introduction

The majority of image sensors used nowadays are composed of CCD and CMOS image sensors [1]. When using these sensors, a simple composition of still images is taken at a predetermined frame rate, to create a video. This approach has several shortcomings; one such problem is data redundancy—each new frame contains information about all pixels, regardless of whether this information is new or not. Handling this unnecessary information wastes memory, power, and bandwidth [2]. This problem might not be critical for the low frames per second (30–60 fps) use case, which is common when the video is intended for human observers. However, applications in computer vision, specifically those that require real-time processing (more so even for high frames per second video), may suffer significantly from such inefficiencies [3].

Several different sensors were suggested to overcome the shortcomings of the frame-based approach, some are bio-inspired (since they can outperform traditional designs [4,5]). Various approaches have been proposed, such as optical flow sensors [6], which produce a vector for each pixel representing the apparent motion captured by that pixel, instead of generating an image, and temporal contrast vision sensors [7], which only capture changes in pixel intensity and utilize hardware distinct from DVS, among others [8,9,10]. However, the overwhelming majority of the market is still CCD and CMOS [1].

Dynamic vision sensors (DVSs) [11] are event-based cameras that provide high dynamic range (DR) and reduce the data rate and response times [2]. Therefore, such sensors are popular these days [12]. Recently, several algorithms have been developed to support them [13,14,15,16,17]. However, current technology is limited, especially in spatial resolution. Each DVS pixel works independently, measuring log light intensity [18]. If it senses a significant enough change (more prominent than a certain threshold), it outputs an event that indicates the pixel’s location and whether the intensity has increased or decreased. The DVS sensor only measures the light intensity and, therefore, does not allow direct color detection. Hence, its vision is binary-like, only describing the polarity of the change in intensity. Due to its compressed data representation, the DVS is very efficient in memory and power consumption [19]. Its advantage is even more prominent when compared to high frames-per-second frame-based cameras. The reduced bandwidth required for recording only changes, rather than whole images, enables continuous recording at high temporal resolution without being restricted to very short videos.

In addition to the color filter array [20], several methods have been proposed to reconstruct color for digital cameras. Examples include optimization-based methods [21,22], image fusion techniques [23], and methods to reconstruct multispectral information [24,25] and 3D Point clouds [26].

Color detection allows more information to be extracted from the DVS sensor and be used in a wide variety of computer vision tasks. These may include extracting DVS events with color, segmenting objects based on color, or tracking various colored objects. We focus on an algorithmic approach to reconstruct color from the binary-like output of the DVS sensor. Unlike current approaches [27], ours does not reduce the native spatial resolution of the sensor. Our algorithm is based on the responses of the DVS sensor to different impulses of an RGB flicker. A list of frames is constructed from the events generated as a response to the flicker, from which features are extracted to reduce redundancy. Two feature extraction methods are described here, and each fits a different reconstruction method. The features are either used as input for a convolutional neural network or a linear estimator, depending on the reconstruction method discussed. The output of these algorithms is a single RGB frame in a standard 8-bit color depth and identical spatial resolution to the input DVS frames; this output is a reconstructed frame of the scene in color. It could then be compared to a frame produced by a traditional camera. Figure 1 shows the workflow presented in this paper. Our contributions are as follows:A fast, real-time, linear method for reconstructing color from a DVS camera.A CNN-based color reconstruction method for DVS.Non-linearity analysis of the DVS-flicker system and an investigation of the DVS response to non-continuous light sources.

## 2. Related Work

In traditional video cameras, color vision is achieved using a simple color filter array (CFA) [20] overlaid on the sensor pixels, with the obvious downside of reducing the spatial resolution of the sensor. In this approach, the output is divided into different channels, one for each filter color. For instance, in the case of the popular Bayer filter, it generally has one red and one blue channel and two green channels for a 2 × 2 binning configuration [28]. These channels can be used directly (without interpolation) in frame-based cameras to produce a colored image or video. One might expect this approach to work just as well for event-based cameras. However, results show that naively composing the different color channels into an RGB image produces results that suffer from high salt-and-pepper type noise and poor color quality (see Figure 4 subfigure C in [29]). A more sophisticated approach to color interpolation from different color channels, such as the ones employed in [29,30,31] (classical algorithms or filters) and [32] (neural network solution) produce better results, especially in terms of noise, but still suffer from poor color quality (see the comparison between [29] and our method in the Result section).

In previous works, the DVS data are assumed to have been produced from a continuous change in lighting. In contrast, in this work, we focus on color reconstruction using the approximate impulse response of a DVS flicker system. Furthermore, the DVS exhibits interesting behavior under non-continuous light changes, which we will discuss in this paper. This was not reported in the literature.

Apart from our model-based approach (based on impulse responses), we also evaluate our method using a CNN model. We chose this approach mainly because of its nonlinearity and spatial correlation, as was demonstrated in previous works.

## 3. Dynamic Vision Sensor (DVS) and Setup

DVS cameras asynchronously output the position of a pixel that experiences a change in logarithmic intensity that is greater than a certain threshold [33]. This method of recording video has several advantages over more traditional synchronous sensors with absolute (as opposed to logarithmic) intensity sensitivity. For example, DVS cameras enjoy higher DR and compressed data acquisition methods, allowing for a more extraordinary ability to detect movement in poorly controlled lighting while using less power, less bandwidth, and better latency.

### 3.1. DVS Operation Method

Pixels in a DVS sensor contain photoreceptors that translate incident photons to the current. The transistors and capacitors are then used to create a differential mechanism, which is activated only when the incident log luminosity difference is greater than a threshold [34].

### 3.2. DVS Response Analysis

One can model the response of the DVS sensor to a change in brightness as a delta function:(1)$$\delta (r-r_{i},t-t_{0})$$
where $r_{i}$ is the pixel (bold because it is a vector) at which the event has occurred, and $t_{0}$ is the time at which the event has occurred. The sensor responds to changes in the logarithmic intensity of each pixel, which can be modeled for a certain pixel at time $t_{k}$, as [12]:(2)$$\Delta L\left(r_{i},t_{k}\right)\geq p_{k}C$$
where L≡log(I), and
(3)$$\Delta L\left(r_{i},t_{k}\right)\equiv L\left(r_{i},t_{k}\right)-L\left(r_{i},t_{k}-\Delta t_{k}\right)$$Here, $I(r_{i},t_{k})$ is the intensity, $p_{k}$ is the polarity of the pixel, which is +1 for an increase or −1 for a decrease in the brightness of that pixel. Variable C corresponds to the threshold that allows a response to be observed and is derived from the pixel bias currents in the sensor.

### 3.3. Creating Video from Bitstream

The DVS outputs a bitstream using the address–event representation (AER); each event detected by the DVS is characterized by an address (indicating the position of the pixel that detects the change), polarity (assigning ‘one’ if the detected change was an increase in brightness and −1 if it was a decrease), and the timestamp of the event detection. In order to turn this list of events into a video, we first choose the desired FPS (we opted for 600 for optimal performance with our specific DVS model, but it is possible to work with up to 1000 fps, and newer models can even go higher). After that choice, we uniformly quantize time. To create a frame, we sum all the events that occurred in each time slice to a single frame, which retains the data about the total event count per pixel during a time period reciprocal to the fps. A similar temporal binning procedure was introduced in [29].

### 3.4. System Setup

We used Samsung DVS Gen3 [34] and positioned it such that it faced a static scene that was 14 inches (35.5 cm) away. For the flicker, we used a screen that was capable of producing light at different wavelengths and intensities. We placed the flicker directly behind the camera, facing the scene, to illuminate it with approximate uniformity, leveraging the fact that its area was larger than the region of the scene captured by the DVS. The flicker changed the emitted color at a 3 Hz frequency. For the calibration process (which will be discussed), we used a Point Grey Grasshopper3 U3 RGB camera, placed adjacent to the DVS (see Figure 2).

The scene was static, so the camera could not see anything if the flicker did not change color. The flicker’s light was reflected off the scene and into the sensor, meaning that if we looked at a single DVS pixel, it measured whether the temporal change in the integral—across all frequencies—of the product of the spectrum, the reflective spectrum of the material, and the quantum efficiency of the sensor surpassed a certain threshold. When light from the source changed in color or intensity, this change was recorded in the DVS.

Using this system, we intend to capture a bitstream generated by the DVS as a response to the flicker, from which a single RGB frame is produced. This frame is an RGB image of the scene with the same resolution as the original DVS video. We present here two algorithmic approaches (one linear and one using CNN, in the next section) for producing that RGB frame. We will create a feature extraction method for each of the two different algorithmic approaches. In order to train the CNN, we will create a labeled dataset using a stereoscopic system with DVS and a standard frame-based camera. Finally, we will provide an explanation of the performances and shortcomings of our method.

## 4. Method—Linear Approach

Here, we introduce a fast, real-time linear method for creating an RGB frame. This method will estimate the color of each pixel based only on the single corresponding DVS pixel. Therefore, our problem is simplified to reconstructing the color of a single pixel from a list of DVS events for that pixel only; we will use the same method for all pixels to create a full RGB image. As will be explained in the following section, we further reduce the problem to estimating the color from a vector of real positive numbers. We generate a few labeled vectors and then build the linear minimum-mean-square-error (LMMSE) estimator using the Moore–Penrose pseudoinverse [35].

### 4.1. Feature Extraction

As presented in Section 3.2, any change in the pixel intensity level causes a change in the sensor event response. When recording the scene for color reconstruction, the intensity and the color of the flicker vary. Therefore, after changing the intensity of light to a new one, the new intensity is treated as if it is a different color. We associate each pixel response with a different activated flicker pulse as an additional feature. In this manner, we assess the features of each pixel for various flicker strengths and colors).

Pre-processing the DVS output yields a list of frames, with each being the sum of events that occurred in a particular time slice. We start by choosing the time intervals, with each corresponding to the response period of the DVS, to a change in the reflected intensity outside the scene, which will later be referred to as integration windows. Thus, a response curve of each pixel to each color change is yielded. We use a flicker that transmits three different colors (RGB) at three different intensities.

In order to reduce the sizes of the data, we use the average response to each color change (a single number per pixel) over a predefined integration window, and responses corresponding to the exact color changes are averaged. The result is a vector of length N+1 (where N=9 is the number of color changes the light source provides; in addition, there is a bias parameter) for each pixel in the 640 by 480 sensor array. To justify this, we will approximate each DVS pixel as an independent LTI system.

#### LTI Approximation

When the flicker is on, each pixel measures the light reflected from a certain part of the scene. Suppose that for a given pixel, this part of the scene is a uniform color. When the flicker is in one color, the pixel will measure the incident photon flux of $F_{i}$; when the flicker changes color (suppose at t=0), the incident photon flux changes to $F_{f}$. Thus, the DVS pixel will output SEVERAL events corresponding to this change (see Figure 3 for example); this is a unique feature resulting from the non-continuous change in light intensity. The number of events depends on the size of the change $\Delta F=F_{f}-F_{i}$. This property is crucial for distinguishing between colors of different brightness levels, such as different shades of gray, since the brightness levels of some colors are directly linked to the intensities of the reflected photons off of them. The stream of events originating from the flicker change lasts well after the flicker transition is over; this means that if the light intensity changes quickly, the DVS will not treat this intensity change as a single event, but rather will continue outputting events for some time, such that the number of events is proportional in some way to the change in intensity.

Suppose that the logarithm of the transmitted intensity of the flicker could be described as a step function:(4)$$f\left(t\right)=au\left(t\right)+u_{0}$$The output current of a DVS pixel is approximated as:(5)$$I_{DVS}\left(t\right)\propto ae^{-bt}$$The exponential form of the output current is implied from the discharge currents of the capacitors in each DVS pixel. We assume the probability of a DVS pixel to register that an event is proportional to the output current:(6)$$f_{DVS}\left(t\right)\propto I_{DVS}\left(t\right)$$As long as $I_{DVS}\left(t\right)$ is above a certain threshold, the DVS sensor will only record events when the log intensity change is sufficiently significant.

The LTI approximation suggests that a sufficient description of the DVS response to the flicker change is to take the number of events that occurred during the flicker change, i.e., integrating over time. Therefore, we suggest the following criterion for characterizing the DVS response to the flicker change:(7)$$F_{\lambda_{1}\to \lambda_{2}}\approx \int_{0}^{\infty }f_{DVS}\left(t\right)dt\approx \int_{0}^{\tau }f_{DVS}\left(t\right)dt$$

We clip the integration at a finite time τ, chosen empirically, in order to approximate the integral. It is approximated as the time difference between the instance of flicker change and the moment when $f_{DVS}\left(t\right)$ has decayed enough. In this work, these moments are identified by measuring the event count of the DVS pixels over an entire frame, with respect to time. A flicker change triggers a local maximum in the event count over time, which then decays proportionally to $f_{DVS}\left(t\right)$. An integration window τ is taken from the frame where the event count is at a local maximum to the frame at which the total event count equals the average event count over all frames. Noise and scattering are taken into account when determining the integration windows.

A pixel of a particular color reflects different amounts of the incident’s photons transmitted by the flicker, depending on their respective wavelengths. Thus, changing the flicker’s color causes the object that the pixel represents to reflect a different amount of light, and the observed intensity changes. This causes a reading of the DVS sensor. Based on these readings, we suggest that it is possible to reconstruct the RGB color profile of the observed scene.

The LTI integration approximation justifies using the LHS of Equation (Equation 7) as the components of the feature vector; however, if we take noise into account, this approximation begins to break down, and the use of spatial correlation is needed for improved approximation.

The exponential form of the DVS event count can be seen in Figure 4, which supports our model of the DVS pixel response probability $f_{DVS}\left(t\right)$. The noisy response shown in Figure 4 is a result of light scattering in addition to the inherent noise in the sensor.

## 5. Method-CNN Approach

The linear approach suffers from low SNR and low color fidelity. The shortcomings of the linear estimator are due to several factors, but mainly the sensor noise (thermal noise is significant for this sensor because it has a large surface area compared to modern CMOS or DSLR sensors) and inherent nonlinearities of the device, which were ignored in the last section’s analysis. The sensor in this work has a thermal noise of 0.03 events/pix/s at 25 °C [12]. An additional form of noise is shot noise, which is significant under low ambient light conditions, such as in our setup. The low color fidelity of the linear approach suggests that the LTI assumption falls short in yielding a highly accurate color reconstruction. Thus, a different, more robust method should be employed for this problem. A natural solution is a convolutional neural network, as is common in image processing settings. This method uses a non-linear estimator for the color; in addition, it takes into account the inherent spatial correlation in the DVS output to reduce output noise.

The input to the network is 288 frames (32 frames for each of the nine flicker transitions) from the DVS, selected to contain the most information about the flicker transition. Since the network is fully convolutional, different spatial resolutions can be used for the input, but the output must be of an appropriate size.

Looking at the output of the DVS, a few things are clear. First, it is sparse; second, it is noisy; and third, there are a lot of spatial and temporal correlations between pixels. This matches previous findings regarding the DVS event stream [12]. In addition, the linear approximation has problems distinguishing between different shades of gray (among other problems); using the spatial and temporal correlations of the data will help produce better results and deal with a certain design flaw of the sensor. It seems that certain pixels respond because neighboring pixels respond, not because they sense a change in photon flux. This is non-linear behavior that cannot be accounted for using a simple linear approximation (see Figure A1 in Appendix A).

### 5.1. CNN Architecture

The network architecture is fully convolutional and inspired by U-Net [36] and Xception [37]. Similar to U-Net, this network consists of a contracting path and an expanding path. However, it also includes several layers connecting the contracting and expanding parts that are used to add more weights that improve the model.

Each layer in the contracting path reduces the spatial dimensions and increases the number of channels using repeated Xception layers [37] based on separable convolution. Each layer in the expanding path increases the spatial dimensions and reduces the channels using separable transposed convolutions. In the end, we move back to the desired output size (in our case, it will be the same as the input size), and the channels will be reduced to three channels (one for each RGB color). The path connecting the contracting and expanding layers does not change the data sizes.

### 5.2. Loss Function

The loss function is a weighted average of the MS-SSIM [38] and $L_{1}$ norm.
(8)$$L\left(Y,\hat{Y}\right)=0.8||\hat{Y}-Y{\left|\right|}_{1}+0.2L_{\mathrm{MS}-\mathrm{SSIM}}\left(\hat{Y},Y\right)$$
where $\hat{Y}$ and *Y* represent the reconstructed and real images, respectively. The coefficients of the different components of the loss function were tuned using hyperparameter optimization. Other losses were tested, including the L2 loss and L1 loss, which only compares the hue and saturation (without the lightness) of the images, or only the lightness without the hue and saturation; however, none outperformed the loss we chose.

The SSIM index is a distortion measure that is supposed to more faithfully represent how humans perceive distortions in images, assuming that the way visual perception works depends significantly on extracting structural information from an image. It is helpful for us because the linear approach (despite being noisy and not producing the most accurate colors) seems to be able to produce images that have the same structures as the original ones.

### 5.3. Training

Data labels are acquired using a dual-sensor apparatus composed of DVS and RGB sensors. For the RGB sensor, we used a Point Grey Grasshopper3 U3 camera, with a resolution of 1920 × 1200 (2.3 MP), fps of 163, and 8-bit color depth for each of its 3 color channels. A calibration process yields matching sets of points in each of the sensors using the Harris Corner Detector algorithm, which is then used to calculate a holography that transforms the perspective of the RGB sensor to the perspective of the DVS sensor.

The calibration process assumes that the captured scene is located in a dark room on a plane at a distance of 14${}^{\prime \prime }$ from the DVS sensor. Therefore, training data are taken on 2D scenes to preserve calibration accuracy. Each training sample contains a series of frames, most of which hold the responses of the scene to the changes in the flicker, and a minority of the frames are background noise frames before the changes in the flicker. For example, in the case of an RGB flicker with three intensities, we use 32 frames per color and intensity, totaling 288 frames.

## 6. Experimental Results

### 6.1. Linear Approach

Some linear reconstructions are shown in Figure 5: The linear approach is shown to reconstruct color, although it is very noisy. This is the result of the pixel-wise approach implemented here, where the spatial correlation between the colors of neighboring pixels is ignored. The fact that noise causes neighboring pixels to experience different event readout patterns causes neighboring pixels to have different feature vectors. Therefore, their reconstructed colors are not similar. This can be rectified by considering the spatial correlation between neighboring pixels, as the CNN approach does.

We also checked the criticality of transmitting light at multiple intensities for distinguishing between various shades of colors, by solely recording the DVS responses to three flicker changes. The result is shown in Figure 6.

Figure 5 shows that using three intensities enables the differentiation between the gray colors in the X-Rite color matrix. However, as seen in Figure 6, using a flicker that transmits RGB light in a single intensity makes the gray colors almost indistinguishable, and is detrimental to the accuracy of the reconstruction. All gray colors have RGB values proportional to (1,1,1), i.e., a gray color has the same value in each RGB channel, indicating the proportion coefficient, or the intensity. Since the DVS output is binary in essence, different gray colors on the scene will respond similarly to the color changes in a flicker with a single intensity. Training with such a setup will result in intensity-mismatched reconstructions. As seen in Figure 3, the number of events recorded depends on the intensity of the flicker relative to the intensity of the scene. Therefore, one can quantify the relative intensity of each pixel by recording its responses to different flicker intensities. Thus, multiple intensities allow us to mitigate the aforementioned problem by measuring the relative intensity of each actual pixel on the scene to the projected intensity of the flicker.

### 6.2. CNN Approach

Figure 7 shows some of our color reconstruction results on 3D scenes that are 14${}^{\prime \prime }$ away from the DVS sensor (distance is measured from the DVS lens to the center of the 3D scene). The resulting reconstruction shows high fidelity and low noise. Thus, this solution is viable for the color reconstruction of 3D and 2D scenes. Small details might not be resolved by this approach due to the smoothing effect of the CNN.

#### 6.2.1. Robustness

Training data have been captured under fixed lab conditions; in particular, the captured scene is at a distance of 14${}^{\prime \prime }$ from the DVS sensor, and the scene is situated in a dark room. Therefore, the robustness of the system to changes in these conditions is shown.

The robustness to changes in the ambient brightness has been studied by using a light source behind the flicker, directed at the scene, which is an extended color matrix. The flicker obstructs some of the incident light from the light source and, therefore, regional intensity variance in the scene has been observed. The light intensity at the scene plane is taken as the average intensity measured at the corners of the color matrix. As seen in Figure 8, brighter ambient light causes worse color reconstruction. The saturation of the scene by enough ambient light causes the flickering color and intensity changes to have a less significant change in the incident intensity measured by the DVS pixels and, therefore, fewer events are recorded.

Distance robustness is also studied, where the color matrix was situated in a dark room, with the same ambient brightness as in the training sessions. Measurements were taken, starting at a distance of 14${}^{\prime \prime }$ from the DVS and increasing by 2${}^{\prime \prime }$ increments. As seen in Figure 9, the reconstruction capabilities of the CNN approach are not limited to the fixed training distance but allow for a distance generalization.

#### 6.2.2. Ablation Study

As part of our analysis and comparison between the different presented methods (linear and CNN), we examined the architecture complexity. In order to accomplish this, we trained several variations of the network with different depths on the same data (and the same number of epochs), and we obtained the following results shown in Figure 10.

An additional component that is worthwhile to study involves the use of different intensities for the flicker (we used three colors with three different intensities). A pertinent issue arises: what would happen to the reconstruction if we only used 1 intensity for the flicker? To that end, we trained the same network architecture for two different datasets. The first set utilized a flicker of three colors, each with three different light intensities, and the second contained the same images, but this time used a three-color flicker with a single intensity per color. In Figure 11, we can see that the reconstruction failed to produce the right colors.

As shown in Figure 12, comparing our results to those of [29], one can see that our method is superior in terms of color reconstruction accuracy, at least on the X-Rite color matrix. The perceived colors of the X-Rite color matrix and the respective MSE values are calculated using 4-point averages of the images from [29] and our reconstruction of the same color matrix. In addition, in contrast to color filter array (CFA)-based color reconstructions, our methods keep the spatial resolutions of the resulting reconstructions close to the native resolution of the DVS sensor.

The method presented in [29] uses DAVIS (dynamic and active vision sensor) pixels with CFA, which means it uses more sensors to create the same resolution pixel as in our method.

## 7. Discussion

The system described in this work manages to create a colored image using the protocol that we presented. However, it has its limitations: the reconstruction accuracy depends heavily on ambient light conditions; if different ambient light conditions are taken into account during training, the reconstruction will be less sensitive to ambient light conditions; finally, the flicker’s intensity and dynamic range affect the reconstruction quality. That being said, the system has the ability to reconstruct color from images outside the training data, as can be seen in Figure 7. One can overcome the latter limitation by changing the intensity levels of the flicker. In addition, our system does not allow the color reconstruction of a nonstationary scene since the response time of the DVS pixels to a flicker color change is nonzero. Therefore, the time it takes for a flicker cycle to be performed limits the timescale of changes in the scene using our approach. The critical distinction lies between events generated by flicker changes and those prompted by movements within the scene. An optical flow algorithm combined with a CNN can be used to overcome this limitation.

Using a DAVIS sensor would also most likely improve the performance and ease the calibration method (the active pixels could be used to create a regular still image used for calibration). In addition, since the DVS has no sense of absolute light intensity (only relative), using a DAVIS sensor could improve the performances of different ambient light conditions.

## 8. Conclusions

This paper presents a novel approach for reconstructing colors from a dynamic vision sensor (DVS) using an RGB flicker. The flicker’s color is changed in a non-continuous fashion, allowing us to determine the spectral data of the scene. Two methods for reconstruction were proposed, with the convolutional neural network (CNN) approach having higher fidelity and less graininess than the linear method. This implementation outperforms a color filter array (CFA)-based reconstruction since the latter reduces the spatial resolution and only samples wavelengths at a few specific wavelengths, while a light flicker allows for sampling wavelengths anywhere on the optical spectrum without a decrease in the spatial resolution. As a future direction, we suggest extending the algorithm to diffuse the color information of pixels over time, for example, by utilizing optical flow maps. To encourage future research we share our code on GitHub.

## Figures and Tables

**Figure 1 sensors-23-08327-f001:**
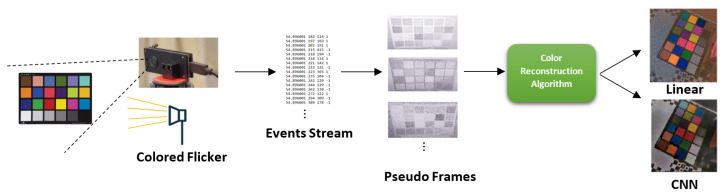
Reconstruction workflow. Left to right: The data are captured with a DVS sensor and a colored light source. Then, an event stream is created from the DVS, which is converted into pseudo-frame representations. Finally, two different color reconstruction approaches can be applied.

**Figure 2 sensors-23-08327-f002:**
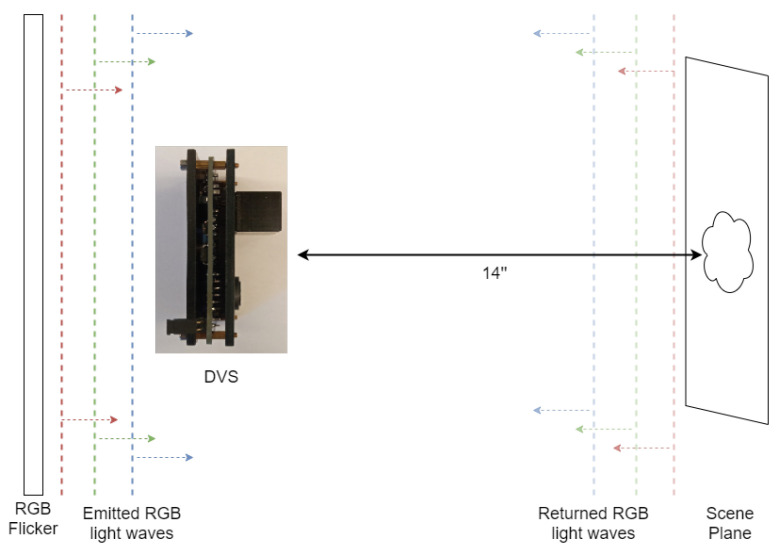
System schematic. The distance from the DVS to the flicker is much shorter than 14${}^{\prime \prime }$. The RGB flicker emits light at a 3 Hz frequency.

**Figure 3 sensors-23-08327-f003:**
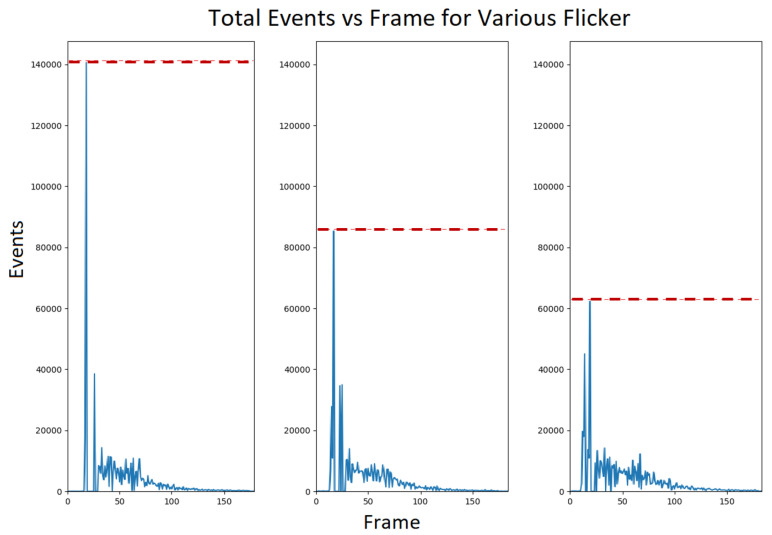
DVS event counts for gray-scale flicker with decreasing gray-scale intensities, left to right. The red dashed lines indicate the peak event count for each gray-scale flicker intensity, showing a correlation between the intensity of the flicker and the number of events recorded.

**Figure 4 sensors-23-08327-f004:**
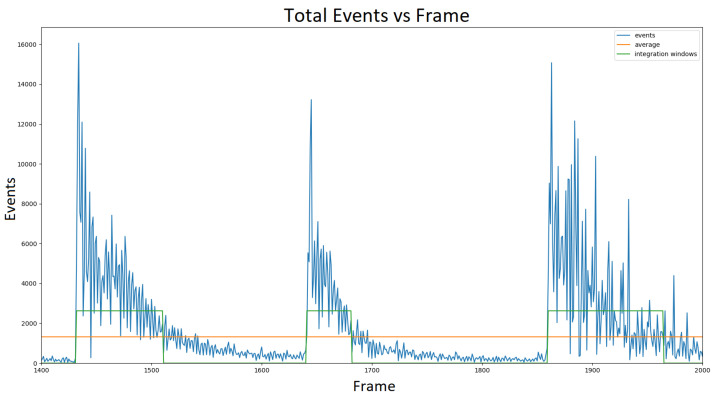
Number of DVS events in a frame vs. frame number. The measurements were recorded while RGB flicker changes took place. The integration windows depict the frames with enough events caused by a flicker color change.

**Figure 5 sensors-23-08327-f005:**
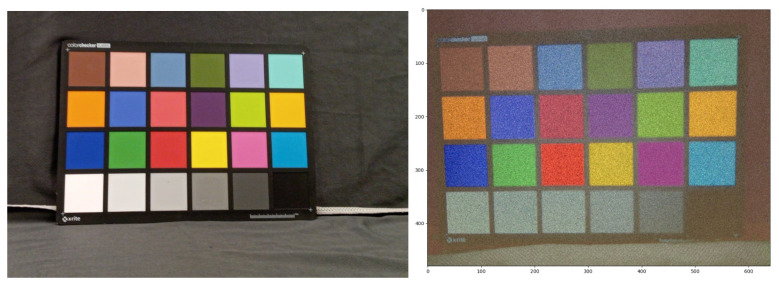
**Left**: X-Rite color matrix. **Right**: color reconstruction using the 9-flicker (3 colors, 3 intensities) linear approach.

**Figure 6 sensors-23-08327-f006:**
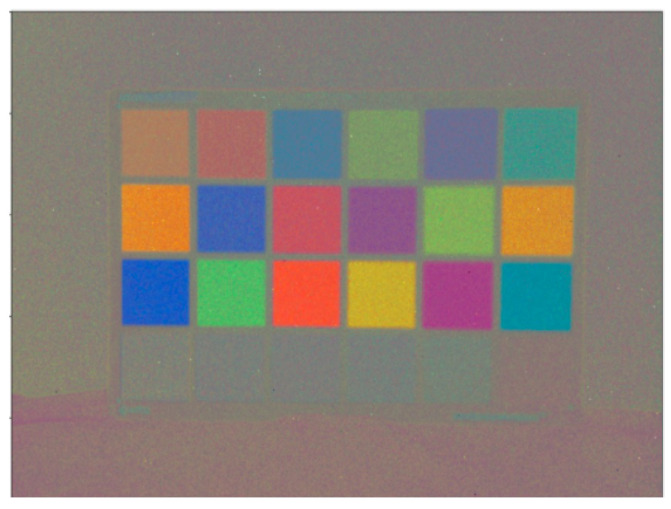
X-Rite matrix reconstruction with a single intensity flicker. The gray colors are almost indistinguishable and the color fidelity has deteriorated, compared to the three-intensity linear reconstruction.

**Figure 7 sensors-23-08327-f007:**
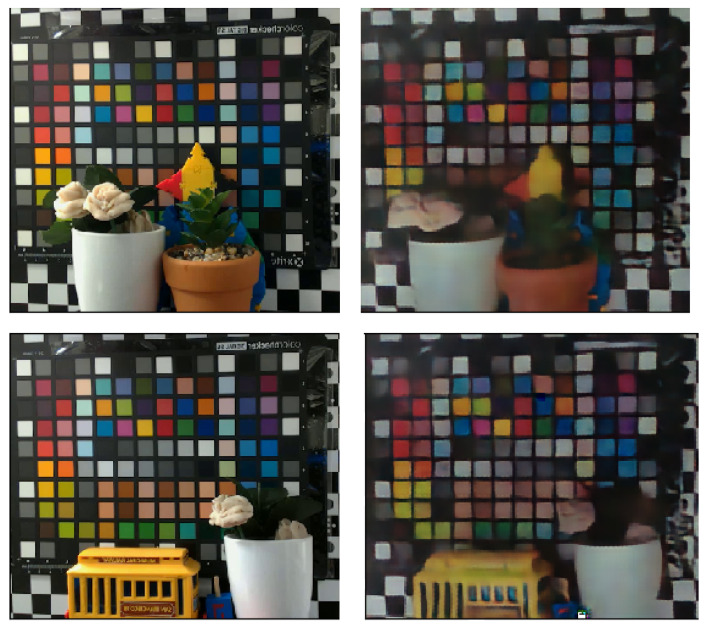
Our model’s reconstruction of the 3D scene. **Left**: The original images, **right**: our CNN-based model reconstructions. The RMSE score for the top right image is 45 and for the bottom right is 47.

**Figure 8 sensors-23-08327-f008:**
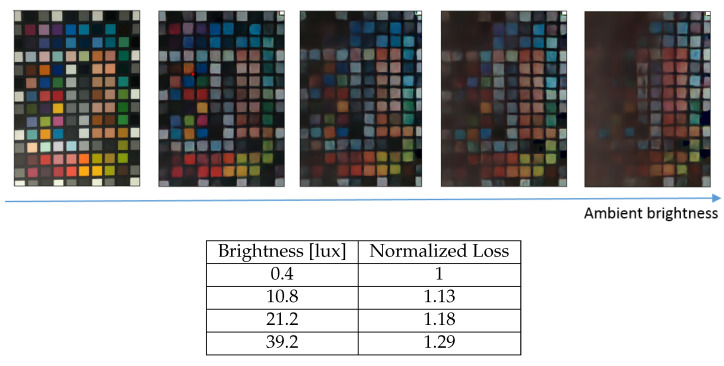
**Top**: Reconstruction for different ambient light conditions. RGB ground truth is shown in the leftmost picture. **Bottom**: relative loss for each reconstruction. The loss is calculated in Equation (Equation 8).

**Figure 9 sensors-23-08327-f009:**
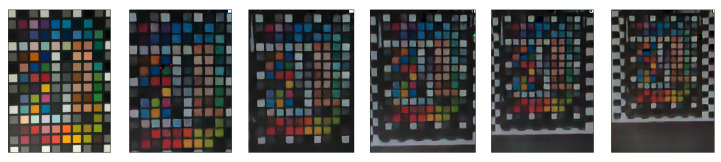
Our CNN-based model reconstruction results for different distances. Distances from left to right: 5.08 cm, 11.176 cm, 17.272 cm, 23.368 cm, 29.464 cm, and 35.56 cm.

**Figure 10 sensors-23-08327-f010:**
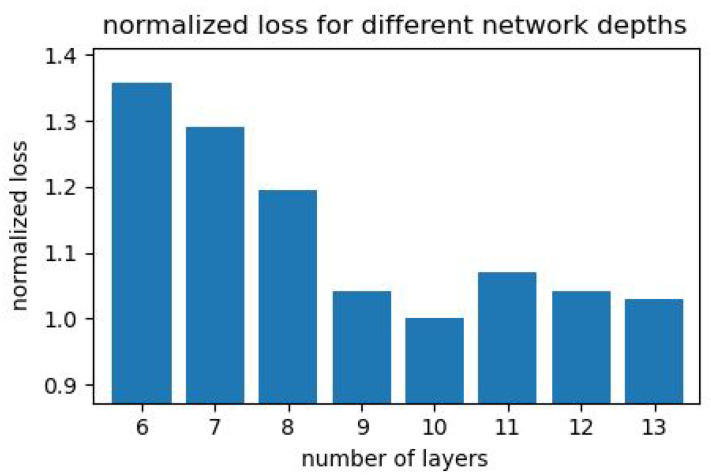
Our neural network normalized the loss across varying numbers of layers. We offer this analysis (without formal proof) as an interpretation of the DVS non-linearity degree.

**Figure 11 sensors-23-08327-f011:**
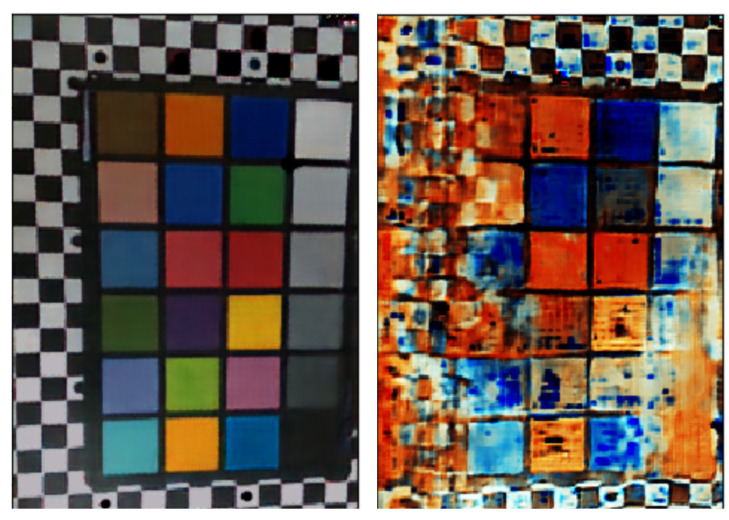
Using the DVS response for different flicker intensity levels significantly improved the reconstruction quality. **Left**: Our CNN-based model reconstruction using 3 different intensities and 3 different colors. **Right**: The same model’s reconstruction using a single intensity and 3 colors.

**Figure 12 sensors-23-08327-f012:**
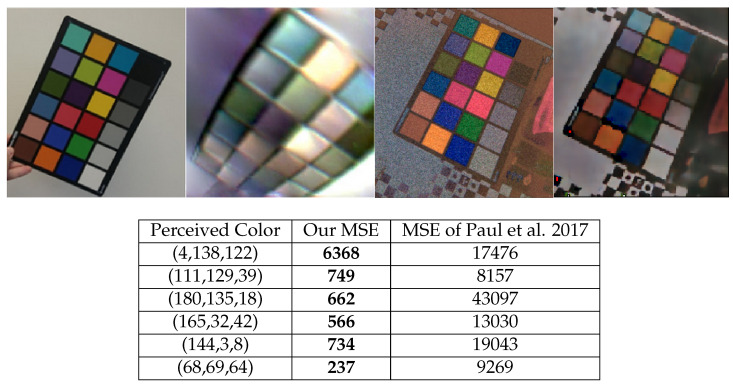
**Left** to **Right**: X-Rite color matrix, reconstruction by [29], our linear reconstruction, our CNN reconstruction.

## Data Availability

The data are reported in the GitHub repository.

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
