# Peer review of "Illumination-Based Color Reconstruction for the Dynamic Vision Sensor"

_sensors, 2023, doi:10.3390/s23198327_

Round 1

Reviewer 1 Report

In this paper, the authors proposed an algorithm to reconstruct color from a binary output of DVS sensor. It exploits the response from multi-channel RGB flickers without reducing the native spatial resolution. Specifically, they extract features based on the LTI approximation and feed the features into CNN.  In summary, It is an innovative paper. Some minor modification need to be implemented:

1 I suggest the author to introduce more about the features extraction part. This part 3.1 seems to have weak connection with previous part. Please add more details to enhance the link with previous contents.

2 The ablation didnt reflect the robustness of the algorithm. I suggest the author add some materials to test the performance of the proposed algorithm with noisy input.

3 I suggest the author to enrich the related work and explain why the CNN is adopted in this algorithm.

Could be further polished.

Author Response

Dear Reviewer,

Thank you for your review.

Please see our answers and modifications according to your comments:

  1. An additional explanation of the feature extraction and its relation to the previous parts has been added to section 3.1. (Please see our addition part in red)
  2. Please note that our presented ablation study focuses on understanding the degree of non-linearity transformation between the input to the systems (delta functions) to the pixel color and intensity. This was done mainly to learn the level of complexity required in the architecture. (Please see our additional part in red)
    The noise robustness of our solution is presented in section 5.2.1 and specifically in Figure 7,8,9 as we’ve analyzed our performance for different distances, different background illumination, and 3D scenes.
  3. An additional explanation about the CCN part and the motivation to use it was added to the related work.

* Comprehensive English correction has been done to the entire paper (not marked in color)

* Major revisions are marked in red/blue/green colors

* Some of the figures were re-arranged

Reviewer 2 Report

This paper uses a DVS sensor and an RGB camera to collect the light intensity change of the reflected light of pixels, and uses the linear method and the CNN method to complete the color reconstruction of the image.

The detailed comments are as follows.

1.     This paper spends too much space introducing DVS. It should focus on color reconstruction algorithms.

2.     In Section 3 and 4, more theoretical analysis and algorithm diagrams are required.

3.     Currently, only stationary scenes are supported in the color reconstruction algorithm. It looks that the application of the algorithm is limited. It is suggested that the authors consider the non-stationary scenes or at least discuss possible direction of improvement in future study.

The presentation is fluent.

Author Response

Dear Reviewer,

Thank you for your review.

Please see our answers and modifications according to your comments:

  1. The reason we focused on how DVS works for two main reasons: First, its operation method is still not widely known by the community, and second, our model-based approach relies heavily on the DVS mechanism, which has to be described in depth in order to understand our solution. Please let us know if you think we should cut some of our background and explanation on the DVS.
  2. The theoretical analysis of our algorithm is presented in section 4 (please see in red our addition to 4.1). Our high-level method is presented in Figure 1 and some in-depth explanation about our algorithm feature extraction is shown in Figures 3+4.
  3. That’s a very correct point, and we indeed relate this in our future work suggestion: “We suggest further extending the algorithm to diffuse the color information of the pixels in time” (please see in green mark in the conclusion). Originally, we aimed in this direction, but we found the color reconstruction task well enough for this project's scope.

* Comprehensive English correction has been done to the entire paper (not marked in color)

* Major revisions are marked in red/blue/green colors

* Some of the figures were re-arranged

Reviewer 3 Report

The authors present a model to reconstruct the colored images via the Dynamic Vision Sensor. The manuscript is focused on an intriguing subject and would be of interest to readers. The overall structure of the paper is acceptable. The findings and data presented in this manuscript are not sufficient. The following must be clarified prior to publication:

1. The entire text of the manuscript must have its English improved. The text has numerous typos and other grammatical mistakes. I would like to suggest that the authors perform a thorough proofreading of the next draft before sending it back in for consideration.

2. The article has an appropriate format and presentation, although there is room for improvement in both areas.

3. The abstract of the paper needs to be polished. You need to discuss in one or two lines how your proposed method overcomes the existing methods and the working phenomena.

4. You should give numbering 1 for the introduction part and then related work at number 2. Therefore, kindly give the correct numbering sequence.

5. Some equations don’t have numbering. Kindly give numbering to equations.

6. The quality of many figures needs to be improved. It seems that you have copied and pasted the figures. Kindly improve the quality of the figures, especially the text in the figures.

7. The proposed method is not explained well, and this work is not sufficient for publication.

8. The simulation results are not enough for this publication.

9. The conclusion of the paper needs improvement.

10. You should compare your work with the latest methods.

11. You should cite some latest papers and check the work of the last 5 years.

The entire text of the manuscript must have its English improved. The text has numerous typos and other grammatical mistakes. I would like to suggest that the authors perform a thorough proofreading of the next draft before sending it back in for consideration.

Author Response

Dear Reviewer,

Thank you for your review.

Please see our answers and modifications according to your comments:

  1. Comprehensive English correction has been done to the entire paper. (not marked in color)
  2. We tried to arrange some of the sections and the figures. We hope it is better now.
  3. Added some explanation to the abstract (please see in blue)

4+5+6. Fixed

  1. Our two methods are explained in detail in sections 4 and 5 (after the definitions and explanation of the DVS in section 3). Do you think we should elaborate more? Can you please specify what is missing?
  2. In our work we have no simulative results, all the presented results are experimental. We developed our methods based on the theory and then examined them in experiments as presented in section 6
  3. We rewrote the entire conclusion section, please see in blue.

10+11. We compared our method and cited the best-known results in the literature presented in the related work section (mainly 2019-2022 and 2017)

* Major revisions are marked in red/blue/green colors

* Some of the figures were re-arranged

Round 2

Reviewer 2 Report

1、Your paper is well-written and informative, but I believe that including more references about color reconstruction from other studies will enhance the credibility of your results.

2、I noticed that the figure 14 is not in the correct location. Please make the necessary changes.

Author Response

Dear Reviewer,

Thank you for your review.

  1. We've added in the introduction some references about color reconstruction from other studies (marked in red)
  2. We arranged the Appendix Figures and added some explanation (marked in red)
  • We've also made some additional English rephrasing along the paper
  • We were asked to relate your suggestion that the paper undergo extensive English revisions: Thank you for the suggestion, we are checking the alternatives on how to do it best.
